



# An experimental study on the aerodynamic loads of a floating offshore wind turbine under imposed motions

Federico Taruffi[1], Felipe Novais[2,3], and Axelle Viré[1]

[1]Delft University of Technology, Faculty of Aerospace Engineering, Kluyerweg 1, 2629 HS Delft, the Netherlands
[2]Politecnico di Milano, Mechanical Engineering Department, Via La Masa 1, 20156, Milano, Italy
[3]Maritime Research Institute Netherlands, Haagsteg 2, 6708 PM, Wageningen, the Netherlands

**Correspondence:** Federico Taruffi (F.Taruffi@tudelft.nl)

**Abstract.** The rotor of a floating wind turbine is subject to complex aerodynamics due to changes in relative wind speeds at the blades and potential local interactions between blade sections and the rotor near-wake. These complex interactions are not yet fully understood. Lab-scale experiments are highly relevant for capturing these phenomena and provide means for the validation of numerical design tools. This paper presents a new wind tunnel experimental setup able to study the aerodynamic response of a wind turbine rotor when subjected to prescribed motions. The present study uses a 1:148 scale model of the DTU 10 MW reference wind turbine mounted on top of a 6 degrees of freedom parallel kinematics robotic platform. Firstly, the thrust variation of the turbine is investigated when single degrees of freedom harmonic motions are imposed by the platform, with surge, pitch and yaw being considered in this study. For reduced frequencies greater than 1.2, it is found that the thrust variation is amplified by up to 150% compared to the quasi-steady value when the turbine is subject to pitch and surge motions, regardless of the amplitude of motion. A similar behaviour is also noticed under yaw motions. Secondly, realistic 6 degrees of freedom motions are imposed by the platform. The motions are derived from FAST simulations performed on the full-scale turbine coupled with the TripleSpar floater and the tests aim at exploring the thrust force dynamics for different sea states and wind conditions, obtaining reasonable agreement with the simulations. Finally, the work shows the capabilities and limitations of an off-the-shelf hexapod to perform hybrid testing of floating offshore wind turbines in wind tunnels.

## 1 Introduction

Nowadays, most offshore wind turbines are installed on support structures that are rigidly mounted on the seabed. However, these fixed foundations are only economically feasible for water depths up to about 60 meters (van Kuik et al., 2016). By contrast, floating offshore wind turbines (FOWTs) that are moored to the seabed unlock the exploitation of wind resources in deeper water. As opposed to bottom-fixed wind turbines, FOWTs are subject to motions in six degrees of freedom that result from the interactions between turbine, wakes, and met-ocean conditions. Due to the complex dynamics involved in such systems, there is a necessity for a comprehensive understanding of the loads on such systems and the suitability of existing numerical models used during the design process.

Scaled model testing is an attractive way to validate numerical models and better understand the physics involved in these systems, at a lower cost than full-scale prototyping. An example of the use of experimental data by the research community is



presented in Robertson et al. (2013), where wave basin data is utilized as a benchmark for code-to-code comparison of fully coupled aero-hydro-elastic engineering tools. Similarly, joint efforts by Bergua et al. (2023), performed comparisons between wind tunnel experiments and numerical models of different fidelities, to validate the aerodynamic loading on a turbine when experiencing large motions caused by a floating support structure.

     In general, wave basin tests have demonstrated great utility in investigating the coupled dynamics of floating offshore wind

turbines. In Goupee et al. (2012), the authors analyzed the response of three different floaters supporting a 1:50 scale model of the NREL 5 MW turbine. As the blades of the model were only geometrically scaled, the rotor was subjected to a thrust force that was significantly lower than expected due to the Froude-scaled low-Reynolds number wind generated at the basin. Hence, a subsequent test campaign was conducted in Goupee et al. (2014), where the rotor blades were re-designed, utilizing an airfoil profile specific for Froude-scaled wind which could match the desired aerodynamic thrust forces. Other endeavors

such as Goupee et al. (2017), Bredmose et al. (2017), Kim et al. (2023) and Meng et al. (2023) included also servo-control capabilities to the model and focused mainly on investigating the global response of the system.

     Even though testing floating offshore wind turbines in a wave basin allows to have both the wind and waves physically present, there are challenges involved mainly when it comes to controlling the wind field and the re-circulation of the flow in the area, as shown by Gueydon et al. (2020). Furthermore, the low Reynolds number encountered when applying Froude

scaling could be detrimental to the aerodynamic response of the turbine, as discussed by Bayati et al. (2018b). Therefore, when it comes to assessing purely the aerodynamic phenomena, it is preferred to perform wind tunnel testing.

     Within the scope of the LIFES50+ project (LIFES50+), a 1:75 model of the DTU 10 MW turbine was designed for wind tunnel testing (Bayati et al., 2016b, a). The model was initially mounted on top of a 2 degrees-of-freedom (DOF) test rig able to impose pitch and surge motions at the base of the turbine. The results of that experiment were utilized to compare the thrust

and torque measurements to the results of a Blade Element Momentum (BEM) model with dynamic wake. Discrepancies were observed between the numerical model and the gathered data, whose causes were not at first clearly identified, and hence motivated further investigation (Bayati et al., 2016c). Later, it was concluded that the outcomes of the experiment were affected by the flexibility of the tower. As a result, a new set of experiments, using a stiffer tower, was conducted. The UNsteady Aerodynamics of FLOating Wind turbines (UNAFLOW) project aimed to understand the aerodynamic response and

wake for a floating offshore wind turbine undergoing large surge motions, the methodology applied for that test campaign is mainly documented in Bayati et al. (2018a), Fontanella et al. (2021) and Mancini et al. (2020). Amidst the test findings, it was observed that the thrust response of the turbine presented a quasi-steady behavior up to reduced frequencies of 0.5 Hz.

     In Fontanella et al. (2022), torque and thrust measurements for a 15 MW wind turbine model were performed for four DOFs. The imposed motion frequencies were defined based on the natural frequencies of each DOF according to two different

floating platforms and the wave spectrum peak for a specific site. Among the findings, it was noticed that the agreement with a quasi-steady model was good for the low-frequency cases, with differences only observed for the pitch. However, at wave frequency, the rotor loads for both surge and pitch were not linearly proportional to the rotor apparent speed, indicating the presence of aerodynamic unsteadiness. In the OC6 project reported by Bergua et al. (2023), both pitch and surge were tested. However, for the range of frequencies considered, the aerodynamic loads presented mostly a quasi-steady response.





In addition to investigating wind turbine loads, a number of wind tunnel tests focus on floating wind turbine wakes and their development under different atmospheric conditions (e.g. Schliffke et al. (2020)). These experimental campaigns typically rely on actuator disc models, with a focus on mid- and far-wake regions rather than turbine loads. Up to date, not many wind tunnel test campaigns with scale model of three-blade floating turbines are documented in the literature and the number of conditions assessed is still scarce. Nevertheless, experimental data is very useful in providing low-uncertainty datasets for the validation of numerical codes. This paper builds on the previous experiments described above whilst extending the motion conditions tested. The turbine investigated here follows the same rotor scaling principles as defined in Bayati et al. (2016b) and is placed on top of a 6 DOF hexapod. Differently from the UNAFLOW project, where only surge was tested, this paper also includes pitch and yaw motions. The aim is to investigate whether the rotor is still subjected to quasi-static loads under these different prescribed motions for a range of motion amplitudes and frequencies. Additionally, prescribed motion time series extracted from 6 DOF fully coupled simulations are also applied to the hexapod and the measured loads at the tower top are compared to the simulated values. It is shown how the present setup is capable of investigating more realistic conditions than previously analysed in the literature.

## 2 Experimental setup

The experimental setup is composed of a wind turbine scale model placed on top of a six degrees-of-freedom hexapod. The setup was tested in the Open Jet Facility (OJF) of Delft University of Technology. The tunnel is a closed-loop open jet test section facility with an octagonal nozzle with a size of $2.85\text{m} \times 2.85\text{m}$ and a contraction ratio of 3:1 opening into a 13m long, 8m high test section. The stream results uniform with a turbulence intensity of $0.5\%$ up to 1m from the nozzle exit and lower than $2\%$ at 6m from the nozzle. The tunnel is powered by a 500kW fan and the flow maximum velocity in the test section is 35m/s. The flow temperature is kept constant by a heat exchanger. A view of the setup in the wind tunnel test section is shown in Fig. 1.

### 2.1 Wind turbine model

The wind turbine model utilized in this work is a 1:148 scale model of the DTU 10 MW reference wind turbine (RWT) (Bak et al., 2013). It is a 3-bladed, fixed-pitch, upwind rotor model designed to operate at a velocity scale of 3. The aerodynamic design was performed in Fontanella et al. (2023). The highlights relevant to the setup description and the analysis of the results are reported here. The blades of the model are not geometrically scaled from the full-scale ones. Instead, to avoid low-Reynolds impaired aerodynamic performance, the rotor is scaled according to a performance scaling oriented at the correct reproduction of the thrust force. The primary objective of reproducing the thrust force is relevant since the model is specifically designed for floating-related testing and the predominant role of thrust in floating offshore wind turbines, especially in the system dynamics and loads, is known (Bayati et al., 2016b). The same performance scaling methodology had been used before for different wind turbine models with different applications and scales (Muggiasca et al., 2021).



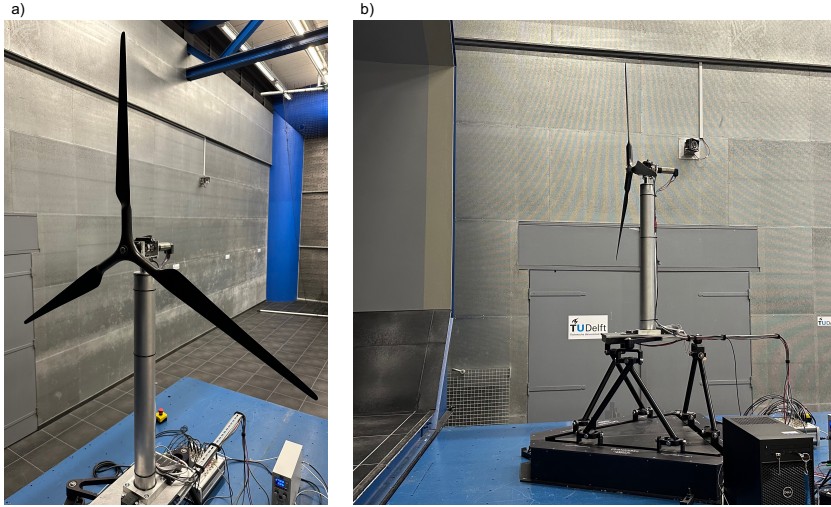

**Figure 1.** View of the experimental setup in the wind tunnel test section: (a) detail of the wind turbine model and (b) complete setup installed in the test section.

A fixed-pitch wind turbine model can be considered beneficial for the scope of the present campaign, given that it excludes any uncertainty on the blade pitch angle. The latter can indeed cause the turbine to operate in a condition different than expected or to operate with different angles among the blades.

The SD7032 airfoil is selected for the design of the wind turbine scaled model. This is a low-thickness profile suitable for low-Reynolds application, as the present one, and is therefore different than the profiles used at full-scale. The static and dynamic polars of the SD7032 airfoil were experimentally characterized by others (LIFES50+). The present work only focuses on unsteady phenomena at rotor scale, rather than at airfoil scale.

The rotor is driven by a servomotor (model Maxon EC-4pole 30 200W) featuring a gearbox (model Maxon GP 32 C 5.8:1) and connected to the rotor shaft by means of an Oldham coupling. The servomotor is speed-controlled by a servo drive (model Maxon Escon 70/10) and is also equipped with a braking resistor (model Maxon DSR 70/30) to dissipate the power generated.

The rotor-nacelle assembly is mounted on a cylindrical aluminium tower which guarantee a high stiffness, important to limit the tower-top deflection and make the rotor follow the desired trajectory precisely. The first fore-aft mode is at around $12.5\text{Hz}$, more than twice the highest motion frequency tested and far from the rotor 1P and 3P frequencies. It's crucial that the tower modes are not excited by any system frequency, to avoid large deflections and limit vibrations that could affect the tests.

## 2.2 Hexapod

The tower base of the scaled wind turbine model described in the previous section is mounted on a parallel kinematics robot capable of imposing motions in 6 degrees of freedom (translations: surge, sway and heave; rotations: roll, pitch and yaw). The motion system used here is the commercially available Quanser Hexapod. The use of a commercial hexapod offers some advantages compared to hand-made stand-alone systems. For example, it gives the possibility of easily recreating the setup in



**Table 1.** Comparison between full scale and model scale operation conditions.

| Parameter | Full Scale | Model Scale | Unit |
|---|---|---|---|
| Cut-in Speed | 4 | 1.33 | m/s |
| Rated Speed | 11.4 | 3.8 | m/s |
| Cut-Out Speed | 25 | 8.33 | m/s |
| Minimum Rotor Speed | 6 | 296 | rpm |
| Maximum Rotor Speed | 9.6 | 473.6 | rpm |
| Rated Thrust | 1619 | 0.012 | kN |
| Rated Torque | 10738 | 0.529 | kNm |

**Table 2.** Main wind turbine model specifications.

| Parameter | Value | Unit |
|---|---|---|
| Rotor diameter | 1.2 | m |
| Hub height | 0.8 | m |
| Blade pitch angle | 0 | deg |
| Tilt angle | 0 | deg |
| Nacelle mass | 1.03 | kg |
| Rotor mass | 0.58 | kg |

different experimental facilities. Additionally, the compactness, lightweight and transportability of the hexapod allow an easy and fast setup installation in the test facility. Tests were performed prior to the experimental campaign in order to assess the capabilities and limitations of the hexapod beyond the specifications and ensure the correct motion tracking in the campaign. The outcome is that the hexapod is capable of operating with the wind turbine at motion frequencies up to 5Hz without amplitude tracking error. The maximum amplitudes are 75mm for translations and 10deg for rotation at low frequencies, which

decrease to 10mm and 1deg at the maximum frequency limited by the maximum velocity and acceleration on each DOF. This corresponds, considering as an example a surge sinusoidal motion case at rated wind, to testing at reduced frequency ($f_r$) of 1.5 with reduced motion amplitude ($A_r$) of 0.01 and normalized velocity variation ($\Delta V^*$) of 0.125. These parameters are further defined in Appendix C. The hexapod motion is commanded by a host-pc and consists of pre-calculated time histories for the various cases to be tested. The pc-host serves as DAQ for the hexapod motion parameters.

## 2.3   Measurements

The measurement system consists of several sensors. A six-component load cell (model ATI mini45 SI-290-10) is placed between the tower top and the nacelle and measures the rotor integral forces and torques. From the load cell measurement, the rotor thrust and torque are obtained. Two MEMS triaxial accelerometers (model TE Connectivity 4030-002-120), placed at the tower base and nacelle locations, measure the translational accelerations. Velocities and position are derived from acceleration



measurements and they are used to enforce motion tracking, as well as to evaluate amplification phenomena that may occur at the rotor location due to tower flexibility concerning the first fore-aft mode in motion cases involving high accelerations. For this scope, the accelerometers are of the low frequency type. The wind turbine operating parameters are retrieved from the motor servo drive and include speed, measured by an encoder (model Maxon HEDL 5540) embedded in the motor, and current, from which the torque can be calculated. All sensor signals are acquired with a sampling frequency of 1000Hz by the

data acquisition (DAQ) system based on a real-time machine (model dSPACE 1302), which combines also a human-machine interface (HMI) for signal visualization and to command the wind turbine operation. Feedback on the hexapod actual motion is also acquired by the DAQ for synchronization purpose. The uniform wind speed is measured by means of a pitot tube installed in the test section and retrieved from the tunnel control system. A schematic representation of the setup is shown in Figure 2.

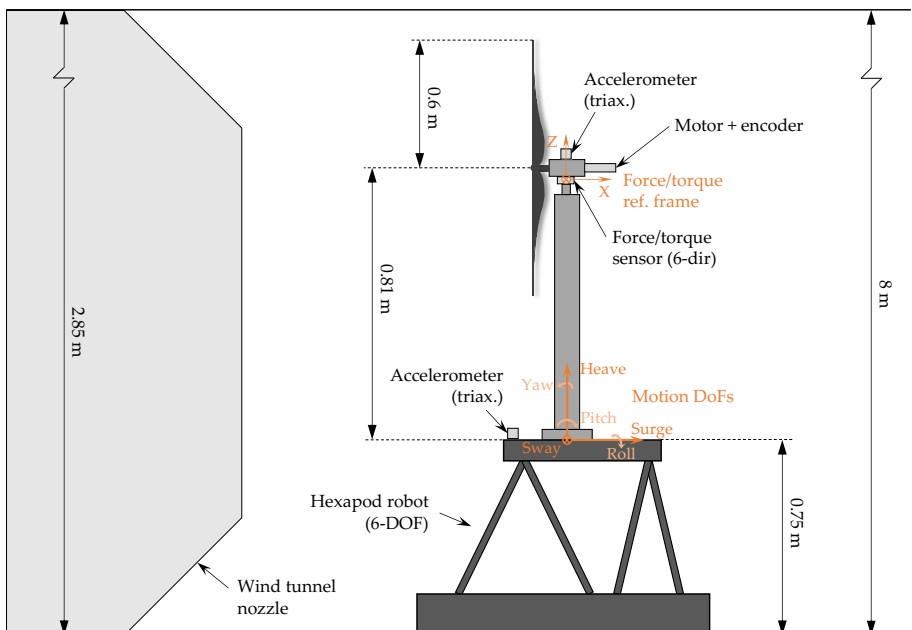

**Figure 2.** Sketch of the experimental setup including motion and force measurement coordinate systems.

## 3 Test matrix

A series of experiments were performed on both a static and a moving turbine. The static tests aim at characterizing the aerodynamics of the scaled rotor, and hence, validating the numerical design of the model. These static tests include the reproduction of the full-scale operating points and the evaluation of the rotor performance in the whole wind speed and rotor speed operating range. In addition, tests aimed at evaluating the aerodynamic sensitivities of the rotor to wind speed and rotor



speed variations were performed around the operating points considered for the moving cases. Regarding the motion tests, the following cases were performed:

- Sinusoidal motion on single DOF for surge, pitch and yaw. The test matrix of surge and pitch motion cases consists of different sets of frequencies and motion velocities. The frequencies are in the range $0.25 - 5$Hz, with a thickening in the high frequencies, and the normalized motion velocities ($\Delta V^*$) are in the range $0.0125 - 0.125$. The motion velocity is identified as a key parameter since the thrust variation, excluding unsteadiness, is directly proportional to it. The motion amplitude is calculated from the velocity and frequency. A similar reasoning applies to the yaw motion case selection. The complete test matrices are displayed in Tables A1, A2 and A3. Two operating points of the wind turbine are considered for these tests, listed in Table 3, representing the rated condition ($U = 4$m/s) and an additional condition in the below-rated region ($U = 2.5$m/s). The duration of each test is different and is such that the tests include $50$ full motion cycles. The main scope of the single DOF tests is to study unsteady aerodynamics effects that may arise at high frequency and assess validity of the quasi-static theory at low-frequency. The values of reduced amplitudes and frequencies for the surge motions considered in this work are shown graphically in Fig. 3, and compared with the values considered in previous experimental studies as reported in Ferreira et al. (2022). As shown, an important contribution of this work lies in having tested higher motion frequencies than in other studies available in the literature.

- Six DOF motions representing the dynamics of a TripleSpar floating support structure. The load cases used as reference were extracted from Krieger et al. (2015) and are relative to a site located in the Gulf of Maine. Wind conditions were adjusted to fall within the range where the DTU 10 MW turbine is operating at a blade pitch angle of $0$deg. Also, no turbulence was used to facilitate the comparison with the wind tunnel, capable of only generating steady wind. The simulations were performed with the FAST model of the TripleSpar floating support structure available in Lemmer et al. (2020). The floater design and numerical model were validated in Bredmose et al. (2017). The turbine is the full-scale reference DTU 10 MW with adjustments to be more representative of the scaled model turbine, namely fully rigid, with a fixed blade pitch and rotational speed. The motion time series were extracted from these fully coupled simulations and converted to model scale, in order to be inputted as prescribed motion in the hexapod motion control system.

For each test run, one or multiple offset tests with a duration of $10$s are run with the Hexapod in the still homing position, the rotor stopped and the wind tunnel off, to calibrate the load cell offset values that are subjected to drifts. Each motion test case is repeated in wind and no-wind conditions. The latter, in which only the motion is activated and both the rotor and the wind tunnel are turned off, is needed to extract the aerodynamic loads from the force measurements, purifying them of the inertial component. The procedure is explained in Section 4.2.

During all the tests with motions, the rotor speed is kept constant. It is important to ensure that the rotor speed variation is as little as possible in order to exclude any effect of it on rotor loads that could be summed to the effect caused by the motion. To ensure good rotor speed tracking, the speed gains in the main motor servo controller are set rather high. The excellent tracking is highlighted considering that the standard deviation in a reference motion case, e.g. a high-frequency surge motion, is around $3.5$rpm while for comparison the standard deviation for a static test is around $3$rpm. A drawback is that this results

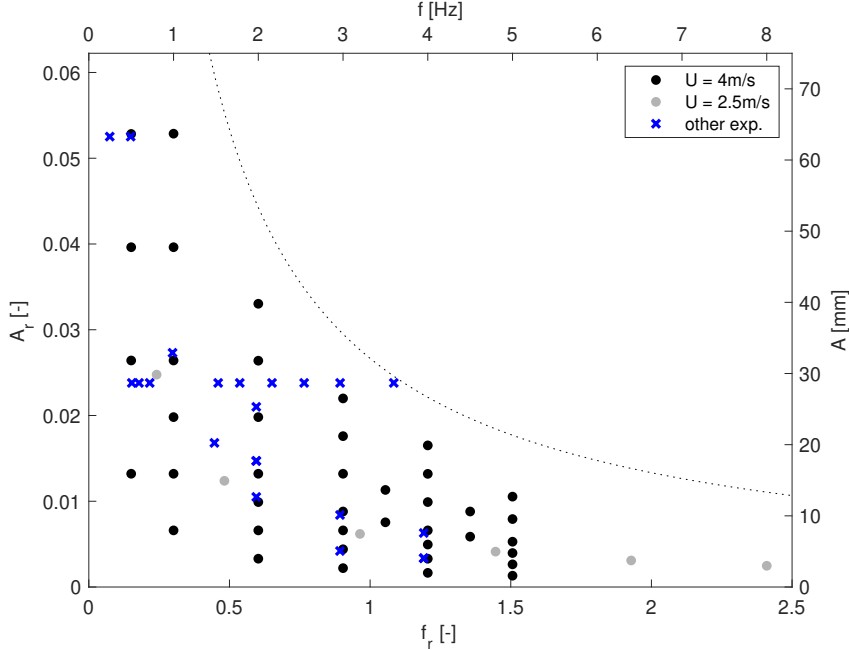

**Figure 3.** Surge motion test matrix visualized according to nondimensional and key operational indicators and compared with previous experimental studies (blue crosses). The cases of the present study are split according to the wind speed in rated ($U = 4\mathrm{m/s}$, black dots) and below-rated ($U = 2.5\mathrm{m/s}$, grey dots). The dimensional parameters (top and left axes) apply only for the present study and rated wind condition.

**Table 3.** Turbine operating parameters for prescribed motion cases.

| Operating Condition | Wind Speed [m/s] | Rotor Speed [rpm] |
|---|---|---|
| Below-rated (BR) | 2.5 | 300 |
| Rated (R) | 4 | 480 |

in a high-varying motor current, and consequently also torque, found in the acquired signals. However, this does not affect the results since they focus on the motion-induced thrust force variation that is not affected by the torque variance but rather

benefits from the very constant rotor speed. An exception is for the steady-state analyses that therefore include averaging.





# 4 Results

## 4.1 Static tests

Prior to performing tests with the turbine under motion, a series of experiments were conducted in order to verify the aerodynamic performance of the scaled rotor. These experimental results were compared to both the reference down-scaled values of the thrust and power of the DTU 10 MW wind turbine and steady-state simulations with the wind turbine model polars utilizing FAST (Jonkman and Jonkman, 2016). Experiments were done twice to assess the repeatability of the measurements and were performed only in the region where the reference turbine, which has a variable-pitch control strategy, would be operating at a pitch angle of 0 degree.

The power curve comparison is shown in Fig. 4. Both torque and thrust measurements are consistent and only a small scattering is observed. The thrust presents a very good match with the FAST simulation results and overlaps the curve of the reference turbine. This is expected since the turbine was specifically designed to operate in that range. For low wind speeds, the experimental thrust measurements are more dispersed and slightly lower than the numerical simulations, which might be due to the wind tunnel's limitation in accurately reproducing flows at such a low wind speed. The torque measurements, both experimental and simulated, indicate lower values compared to the DTU 10 MW turbine. This difference can be attributed to the utilization of a low-Reynolds number airfoil, the SD7032, which is less efficient than the airfoil used in the reference rotor.

Various wind conditions and rotational speeds were also utilized to conduct tests on the turbine. The map in Fig. 5 displays the turbine's thrust coefficient, $C_T$, and power coefficient, $C_P$, across different operating conditions. Additionally, the main operational points considered in this paper are highlighted (red symbols). Finally, Fig. 6 shows the thrust and power coefficients of the main wind speed used in the experiment, $U = 4\mathrm{m/s}$, as a function of the tip-speed ratio.

An uncertainty analysis of the steady-state thrust force at the two main operating points considered for the motion cases (R and BR) was performed. The standard deviation was calculated making use of the average thrust evaluated over the stationary time windows present in each motion test data, resulting in an uncertainty of $9.8\%$ and $10.3\%$ for R and BR operating points.

## 4.2 Thrust variation

When the turbine is set under motion, the force measurement needs to be corrected to subtract the inertial and gravitational forces, in order to isolate the aerodynamic contribution due to the dynamic motions. This is achieved by subtracting the forces measured during no-wind tests from the force measured during wind tests. This work applies this correction to both time and frequency domains, the former for visualization and the latter for quantitative analyses. To evaluate the thrust variation in frequency, the procedure involves cutting the signals to remove transients and analyze only the relevant motion parts, performing a fast Fourier transform (FFT) on the signals, and obtaining complex numbers representing the force measurements for the wind and no-wind tests at the motion frequency. The difference between the complex numbers represents the thrust variation at the motion frequency. However, the phases of the complex numbers need to be corrected for synchronization purposes. This involves subtracting the phase of a corresponding reference signal from the wind and no-wind force complex numbers, identified in the tower-base acceleration measure. The tower-base acceleration signal is preferred over the motion feedback signal

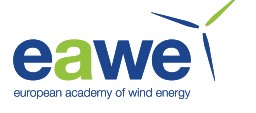


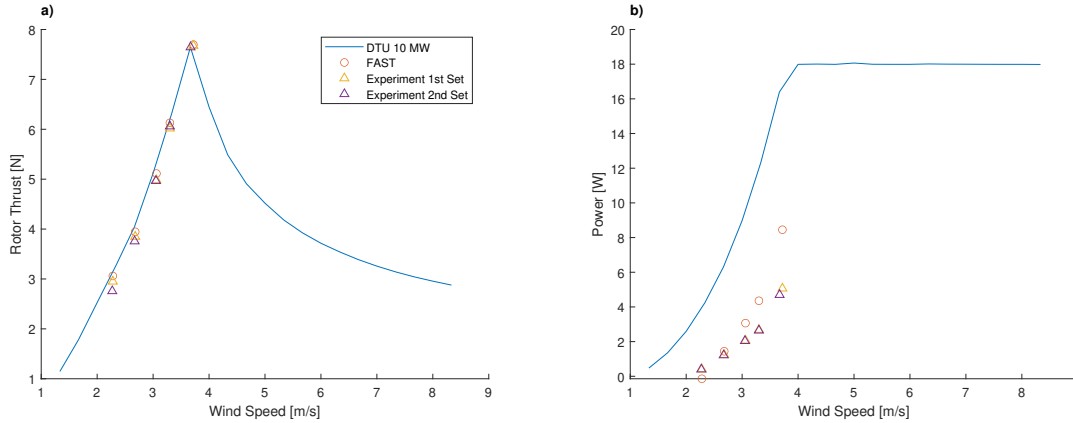

**Figure 4.** Thrust, a), and power, b), comparison of the wind turbine model as a function of the wind velocity.

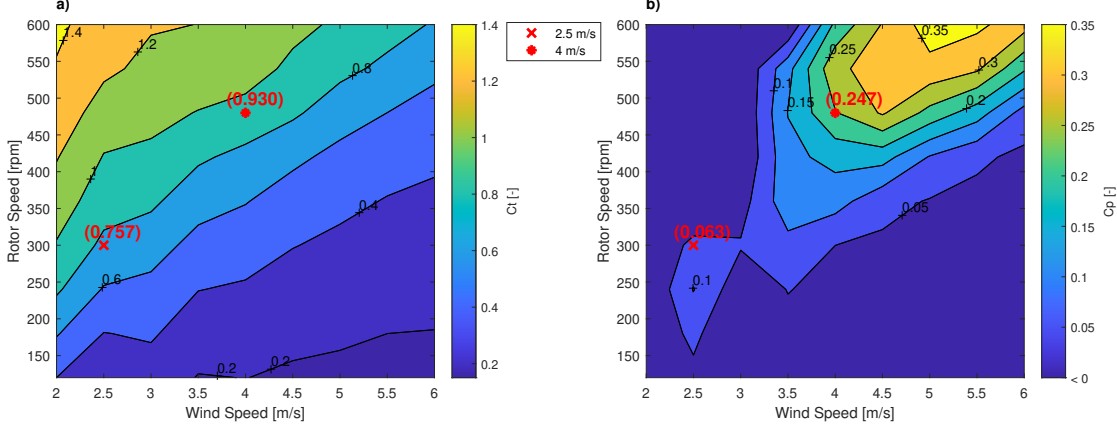

**Figure 5.** Map of the power and thrust coefficients for the turbine at different operation points. $C_T$, a), and $C_P$, b), values utilized for the prescribed motion cases are highlighted by red symbols.

because it is less processed and thus less subject to inconstant delays. It is also preferred over the nacelle acceleration measure
because the latter could be influenced by the aerodynamic loading leading to a possible phase shift of the acceleration signal
in wind tests only, hence harming the synchronization.

The method to calculate the thrust force variation can be summarized as

$$|\Delta T| \, e^{i\phi\Delta T}\Big|_{\hat{f}} = |\Delta F_X^w| \, e^{i\left(\phi\Delta F_X^w - \phi a_{tb,X}^w\right)} - |\Delta F_X^{nw}| \, e^{i\left(\phi\Delta F_X^{nw} - \phi a_{tb,X}^{nw}\right)}\Big|_{\hat{f}} \tag{1}$$

where $|\Delta T|$ is the thrust variation amplitude, $\phi\Delta T$ is the thrust variation phase, $|\Delta F_X^w|$ and $\phi\Delta F_X^w$ are the force variation
amplitude and phase measured in $X$−direction (i.e. the direction of thrust) in wind condition, $|\Delta F_X^{nw}|$ and $\phi\Delta F_X^{nw}$ are the
force variation amplitude and phase measured in $X$−direction in no-wind condition, $\phi a_{tb,X}^w$ and $\phi a_{tb,X}^{nw}$ are the phase of the
acceleration measured in the same direction in wind and no-wind condition and $\hat{f}$ is the frequency of motion.

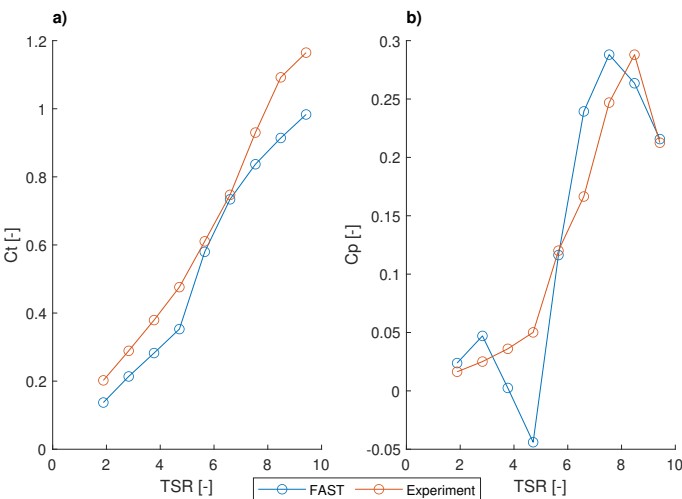

**Figure 6.** $C_T$, a), and $C_P$, b), as a function of the tip-speed ratio for a constant wind speed of 4 m/s.

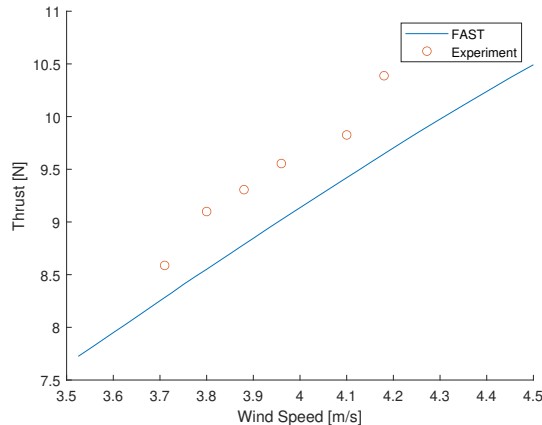

**Figure 7.** Steady-state measurements of the thrust force as a function of the wind speed for a constant rotational speed of 480 rpm.

For surge and pitch single-DOF motion cases, the experimental thrust variation is compared with a quasi-steady estimation based on the quasi-steady theory. This theory, described in Appendix C, calculates the thrust fluctuations resulting from the variation in relative wind speed experienced by the rotor during motion. The comparison with experimental results helps determine the presence or absence of unsteady phenomena. The nominal quasi-static thrust variation is generally calculated from the motion amplitude, as in Eq. C4, but the actual maximum motion velocity may differ. To address this, the velocity in the quasi-static estimation is evaluated from the spectral amplitude of the nacelle velocity, obtained by integrating the acceleration signal in $X-$direction at the motion frequency. Therefore, instead of considering a single value of quasi-steady thrust variation for each set of $\Delta V^*$, the value is evaluated individually for each test case to take into account the actual wind speed variation




that the rotor faces. The estimated quasi-static thrust variation may deviate from the nominal value for two reasons: (i) the amplification effect due to the flexibility of the tower, which would cause the velocity at the hub to increase compared to the nominal value, and (ii) the inability of the Hexapod to track the prescribed motion for particularly high-frequency cases, which would cause the velocity at the hub to decrease. The method adopted here allows to fully take into account the aforementioned

amplification effects. The latter could have induced increases in measured thrust variations at high frequencies that would have been erroneously attributed to unsteady phenomena. In reality, however, it is noticed that the amplification contribution only results in a small part of the total thrust increase, as can be seen for example in Fig. 8. It can be inferred that the remaining part is to due to the increase in frequency only, and thus to the arising of unsteady phenomena. In addition to the aforementioned post-processing, for estimating the nacelle velocity in pitch cases, the gravitational contribution in the measured acceleration

has to be subtracted.

The aerodynamic wind-to-thrust sensitivity ($K_{UT}$), used to calculate the quasi-steady thrust variation according to Eq. C3, is evaluated by linear regression on the outputs of multiple steady-state numerical simulations performed in FAST for wind speeds around the operating points and equals $K_{UT} = 2.85\mathrm{Ns/m}$ for $U = 4\mathrm{m/s}$ and $K_{UT} = 1.58\mathrm{Ns/m}$ for $U = 2.5\mathrm{m/s}$. Here numerical values are preferred over values obtained by specifically performed static tests (results shown in Fig. 7) because of

240 the uncertainty in the wind speed measurement for small velocity variations. The uncertainties in wind speed measurements do not affect the results as the wind speed variation during tests is determined by the motion system while the wind tunnel is set to a steady value. Uncertainties in wind speed measurements only impact static tests, also considering that the aerodynamic sensitivity remains approximately constant in the neighbourhood of the nominal wind speed that is greater than the uncertainty.

### 4.2.1 Surge motion

The thrust variation $\Delta T$ is evidently dependent on frequency. According to the results shown in Fig. 8, the experimental thrust variation amplitude is approximately constant and equal to the quasi-static thrust variation for low frequencies. By contrast, it increases significantly, up to an increment of about $50\%$, for the highest frequencies tested. The threshold frequency above which the thrust variation amplitude starts deviating from the quasi-static estimation is identified to be around $4\mathrm{Hz}$ for rated wind cases ($U = 4\mathrm{m/s}$) and $3\mathrm{Hz}$ for below-rated wind cases ($U = 2.5\mathrm{m/s}$), both corresponding to a reduced frequency of

around 1.2. To our knowledge, this is larger than the highest reduced frequency tested in the literature. For example, Mancini et al. (2020) considered reduced frequencies up to 1.2 and showed increasing scatter of the results at these high frequencies. The authors mentioned that this was likely due to the inception of unsteady effects. This is confirmed in this work for increasing values of $f_r$. The thrust variation phase, shown in Fig. (b), does not show a clear trend. This effect does not appear to be dependent on the motion velocity parameter, $\Delta V$ as it can be noticed in Fig. 9. Some discrepancies that can be seen for the

lowest values of $\Delta V$ are likely to be caused by the small forces to be measured in these cases that increase the uncertainty of the measure The effect is equally noticeable independently of the wind speed, as Fig. 10 shows. However, similarly to the low $\Delta V$ point, for the below-rated wind speed, the forces to be measured are small. Moreover, the uncertainty on the wind speed measurement is greater at lower speeds, possibly affecting the aerodynamic sensitivity value at this operating point.





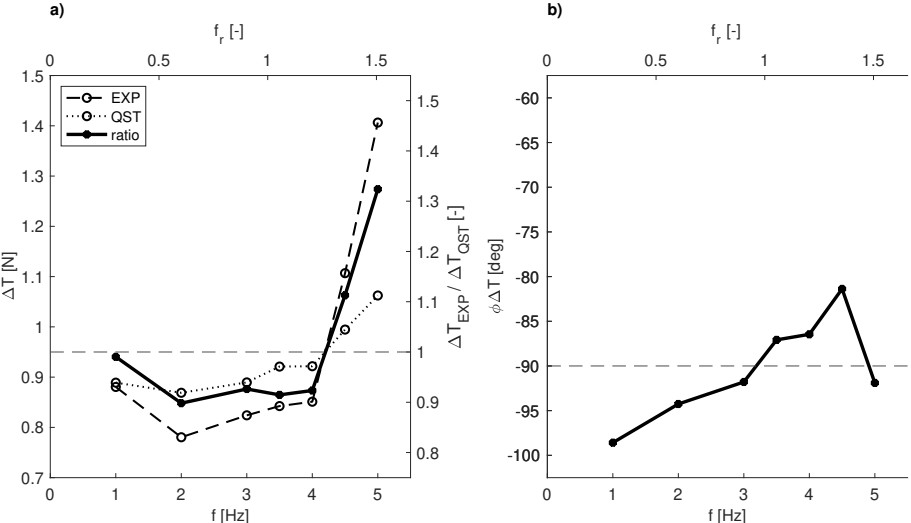

**Figure 8.** Thrust variation as a function of surge motion frequency for the rated wind speed of $U = 4\mathrm{m/s}$ and $\Delta V^* = 0.075$: a) experimental amplitude (dashed), quasi-static amplitude (dotted), ratio between experimental and quasi-static (solid) and b) experimental phase (solid). The horizontal dashed line corresponds to the quasi-static values.

The time history of two surge motion cases, a low-frequency case and a high-frequency one, can be visualized in Fig. 11
in a one-period window showing averaged records of motion and force measures over a 30 motion cycle. The 10 cycles at the beginning and at the end of each record are omitted to avoid any transient effect.

### 4.2.2 Pitch motion

The pitch motion tests reveal similar findings, as shown in Fig. 12 for an illustrative case corresponding to $\Delta V^* = 0.075$. As for the surge case, we observe that the thrust variation amplitude is essentially independent of the frequency of motion
for frequencies up to $4\mathrm{Hz}$. Below that value, the trust variation replicates the quasi-steady value. By constrast, it significantly increases for higher frequencies. Also here the phase does not show a clear trend. The similarity between the surge and pitch results is due to the small rotation involved in the pitch cases and this contributes to strengthening the repeatability of the present findings.

### 4.2.3 Yaw motion

For the cases under yaw motion, no hub horizontal velocity is actuated. Therefore, the results in terms of thrust variation are shown without quasi-static comparison. Results in Fig. 13 still exhibit a similar outcome to the cases with surge and pitch motions. However, the effect is less clear than in previous cases. Here the thrust variation amplitude first decreases in the frequency range $0.5 - 2\mathrm{Hz}$, then slightly increases for frequencies up to $4\mathrm{Hz}$ and then rises more steeply. The phase is fairly constant with frequency and lies between $-105$ and $-95\mathrm{deg}$.



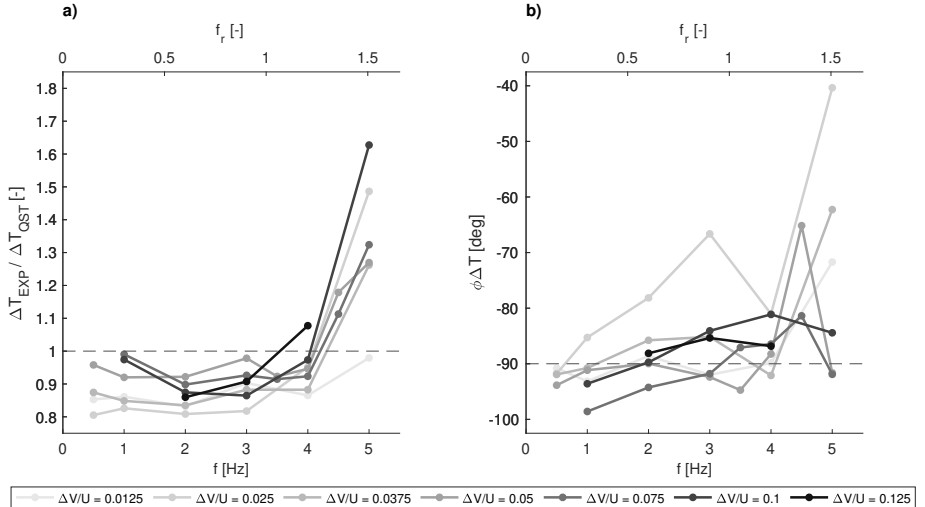

**Figure 9.** Thrust variation as a function of surge motion frequency for different $\Delta V^*$ parameters at the rated wind speed of $U = 4\mathrm{m/s}$: a) ratio between experimental and quasi-static amplitudes and b) experimental phase. The line colour represents different values of $\Delta V^*$ (with increasing values from light grey to black), and the horizontal dashed line corresponds to the quasi-static values.

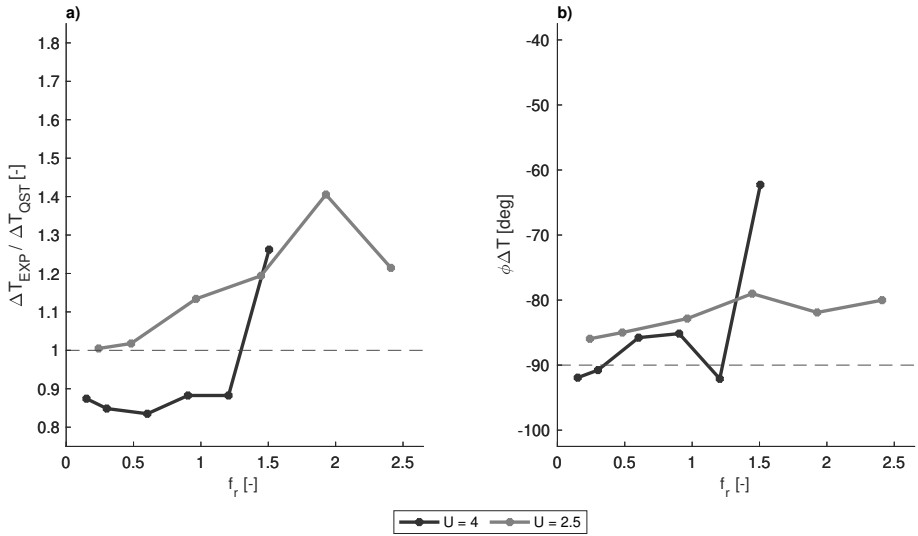

**Figure 10.** Thrust variation as a function of surge motion reduced frequency and wind speed, for $\Delta V^* = 0.0375$: a) ratio between experimental and quasi-static amplitudes and b) experimental phase. The line colour differentiates between a wind speed of $U = 4\mathrm{m/s}$ (black) and $U = 2.5\mathrm{m/s}$ (grey), and the horizontal dashed line corresponds to the quasi-static values.





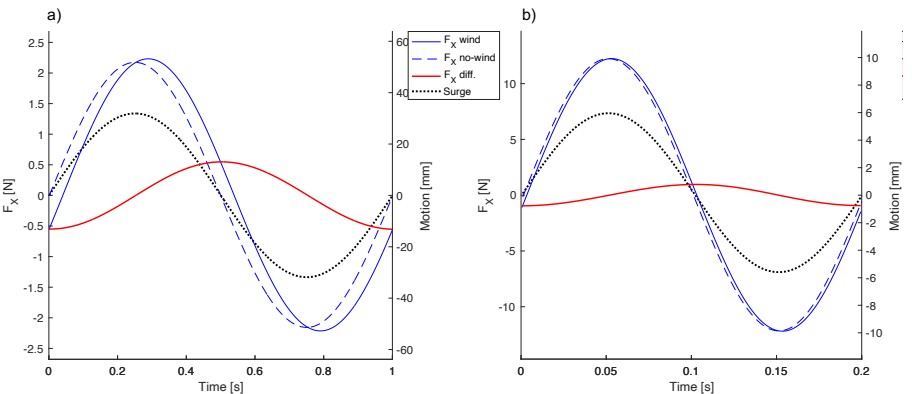

**Figure 11.** 30 cycles averaged records of surge cases: a) motion frequency 1Hz and b) motion frequency 5Hz, parameter $\Delta V^* = 0.05$. The blue lines are the force measured in X direction for the wind case (solid) and no-wind case (dashed), the red line is the difference between them, thus representing the aerodynamic contribution (i.e. the thrust force), and the black dotted line is the surge position. The records are low-pass filtered at 1.5Hz and 7.5Hz, respectively, corresponding to 1.5 times the motion frequency.

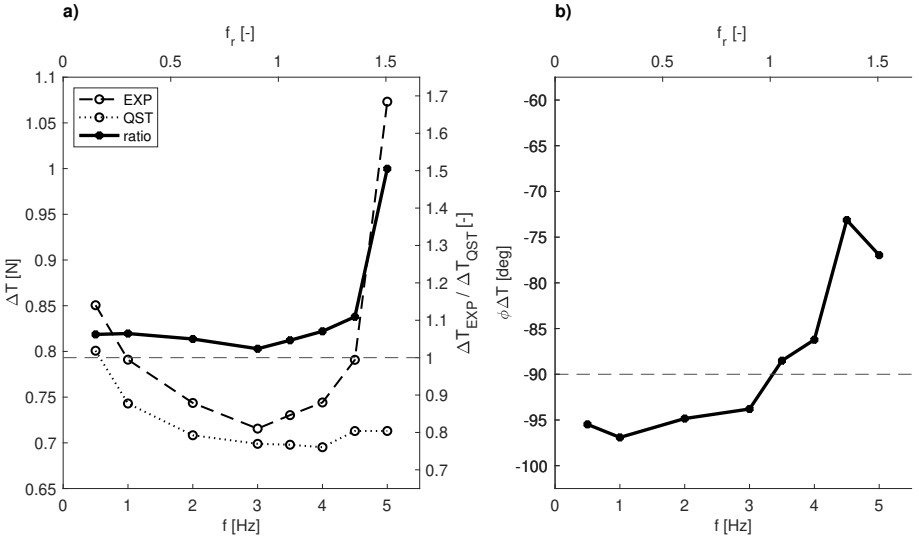

**Figure 12.** Thrust variation as a function of pitch motion frequency for the rated wind speed of $U = 4\mathrm{m/s}$ and $\Delta V^* = 0.075$: a) experimental amplitude (dashed), quasi-static amplitude (dotted), ratio between experimental and quasi-static (solid), and b) experimental phase (solid). In both plots, the horizontal dashed line corresponds to the quasi-static values.

### 4.3 Wave load cases

For wave cases, in which the turbine model is moved according to a prescribed 6 DOF time history of floater motion for different wind-wave load cases, the analysis focuses on the thrust force variation induced by the realistic motion. The outcome

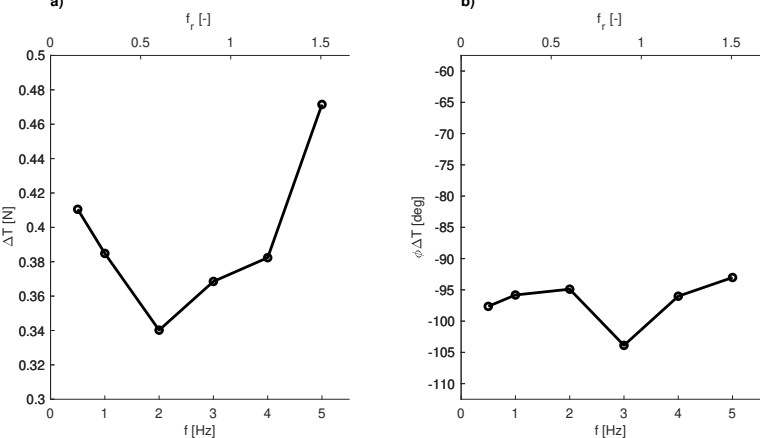

**Figure 13.** Thrust variation as a function of yaw motion frequency for the rated wind speed of $U = 4\mathrm{m/s}$ and the same value of motion velocity of $19\mathrm{deg/s}$: a) experimental amplitude and b) phase.

of the frequency analysis is presented in Fig. 14, which shows the spectra of measured thrust for two load cases. The results are compared with the values predicted by the FAST simulations on the full-scale FOWT system used to produce the motion

time histories. The two load cases, whose parameters are reported in Table B1, represent two cases for a normal sea state and with below-rated and rated wind conditions, respectively.

The comparison shows a good agreement up to the wave frequency range $(0.05 - 0.2\mathrm{Hz})$. For higher frequencies, the experimental spectrum deviates from the numerical one and is higher and flat mainly due to the presence of noise, considering that at model scale they correspond to frequencies greater than $10\mathrm{Hz}$ which are outside the scope of study being above the limits of

the Hexapod. The highest peak that is visible in both experimental spectra corresponds to the 1-per-turn (1P) frequency of the rotor.

The agreement is good unless for the harsher sea states that, causing higher motions, fall beyond the limits of the Hexapod that is incapable of correctly tracking the tabulated motion. However, those cases represent only the more severe sea states.

## 5 Conclusions

In this study, a new experimental setup is developed to analyze the aerodynamics of a scaled wind turbine under prescribed floating motions. The setup uses a scaled version of the DTU 10 MW reference wind turbine that has been extensively studied and is suitable for floating wind applications. Since wind turbine control is outside the scope of the present work, the scaled model used here has a fixed blade pitch, which simplifies the model and avoids any additional uncertainties in the experimental results. The motions are prescribed using a commercially-available hexapod that can prescribe motions in 6

degrees-of-freedom. The present work distinguishes itself from other studies in the literature in multiple ways. Firstly, both single and realistic 6-DOF simulated motions are investigated here, whilst to our knowledge, only single and coupled two DOF

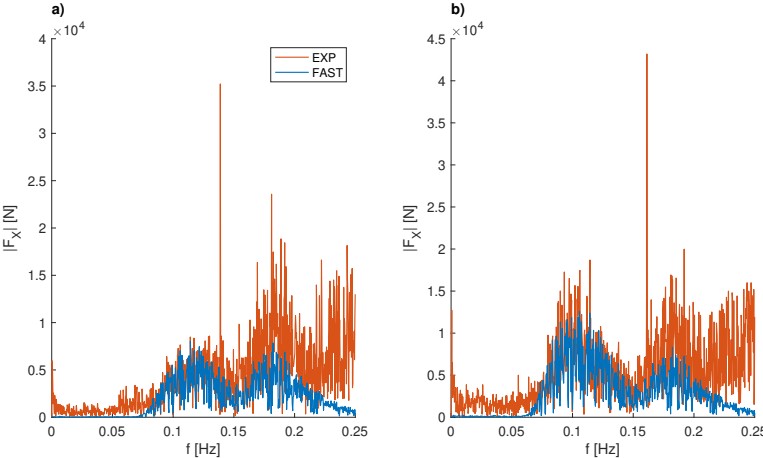

**Figure 14.** Spectral amplitude of thrust force for case LC3 (a) and case LC4 (b): experimental measure (orange) vs. FAST simulation output (blue). The comparison is made at full-scale.

have been investigated in the literature so far. Secondly, the range of imposed motion amplitudes and frequencies surpasses what is currently available in the literature.

The results obtained from the single DOF motion experiments clearly indicate the presence of unsteady effects. It was observed that the variation in thrust force increases as the reduced frequencies exceed about 1.2. This signifies that the thrust force experiences greater fluctuations under higher reduced frequencies. However, when the reduced frequency falls below 1.2, the thrust variation remains constant, implying that the quasi-static theory can be applied in such cases. No discernible effect of wind speed and motion velocity on thrust amplification was evident from the experimental data. It was also shown that the present experimental setup is capable of reproducing the full-scale behaviour of a floating wind turbine under more realistic met-ocean conditions. In particular, reasonable agreement was obtained between the experimental thrust force and that computed through FAST simulations for a TripleStar floating wind turbine under normal sea state conditions, the only discrepancies occurring at large frequencies that fall outside the main wave frequency range.

These findings contribute to a deeper understanding of the aerodynamic behavior of floating offshore wind turbines, and the observed unsteady effects underscore the importance of considering them in the design and analysis of floating wind turbines. By expanding the range of investigated parameters and pushing the boundaries of previous studies, this experimental campaign provides valuable insights into the loading on floating wind turbine rotors and can be used to further validate numerical models.

The observations outlined in this paper provide a compelling reason to pursue additional research into the impact of floater motions on wind turbine performance. To date, experiments have primarily concentrated on investigating one or two degrees of freedom. Nevertheless, operational floating offshore wind turbines experience combined motion across all six degrees of freedom. Hence, additional research in this field could focus on exploring the impact that the coupling of these degrees of freedom has on the aerodynamic response of the turbine.





## Appendix A: Single Degrees-of-Freedom Prescribed Motion Test Matrix

**Table A1.** Test matrix for the prescribed surge motion cases.

| Surge - 2.5 m/s, 300 rpm | | | | | Surge - 4 m/s, 480 rpm | | | |
|---|---|---|---|---|---|---|---|---|
| $\Delta V^*$ | f [Hz] | A [mm] | $f_r$ [-] | | $\Delta V^*$ | f [Hz] | A [mm] | $f_r$ [-] |
| 0.0325 | 0.5 | 29.8 | 0.24 | | 0.0125 | 0.5 | 15.9 | 0.15 |
| 0.0325 | 1 | 14.9 | 0.48 | | 0.0125 | 1 | 8 | 0.3 |
| 0.0325 | 2 | 7.5 | 0.96 | | 0.0125 | 2 | 4 | 0.6 |
| 0.0325 | 3 | 5 | 1.44 | | 0.0125 | 3 | 2.7 | 0.9 |
| 0.0325 | 4 | 3.7 | 1.92 | | 0.0125 | 4 | 2 | 1.2 |
| 0.0325 | 5 | 3 | 2.4 | | 0.0125 | 5 | 1.6 | 1.5 |
| 0.05 | 0.5 | 39.8 | 0.24 | | 0.025 | 0.5 | 31.8 | 0.15 |
| 0.05 | 1 | 19.9 | 0.48 | | 0.025 | 1 | 15.9 | 0.3 |
| 0.05 | 2 | 9.9 | 0.96 | | 0.025 | 2 | 8 | 0.6 |
| 0.05 | 3 | 6.6 | 1.44 | | 0.025 | 3 | 5.3 | 0.9 |
| 0.075 | 0.5 | 59.7 | 0.24 | | 0.025 | 4 | 4 | 1.2 |
| 0.075 | 1 | 29.8 | 0.48 | | 0.025 | 5 | 3.2 | 1.5 |
| 0.075 | 2 | 14.9 | 0.96 | | 0.035 | 0.5 | 47.7 | 0.15 |
| | | | | | 0.035 | 1 | 23.9 | 0.3 |
| | | | | | 0.035 | 2 | 11.9 | 0.6 |
| | | | | | 0.035 | 3 | 8 | 0.9 |
| | | | | | 0.035 | 4 | 6 | 1.2 |
| | | | | | 0.035 | 5 | 4.8 | 1.5 |
| | | | | | 0.05 | 0.5 | 63.7 | 0.15 |
| | | | | | 0.05 | 1 | 31.8 | 0.3 |
| | | | | | 0.05 | 2 | 15.9 | 0.6 |
| | | | | | 0.05 | 3 | 10.6 | 0.9 |
| | | | | | 0.05 | 3.5 | 9.1 | 1.05 |
| | | | | | 0.05 | 4 | 8 | 1.2 |
| | | | | | 0.05 | 4.5 | 7.1 | 1.35 |
| | | | | | 0.05 | 5 | 6.4 | 1.5 |
| | | | | | 0.075 | 0.5 | 95.5 | 0.15 |
| | | | | | 0.075 | 1 | 47.7 | 0.3 |
| | | | | | 0.075 | 2 | 23.9 | 0.6 |
| | | | | | 0.075 | 3 | 15.9 | 0.9 |
| | | | | | 0.075 | 3.5 | 13.6 | 1.05 |
| | | | | | 0.075 | 4 | 11.9 | 1.2 |
| | | | | | 0.075 | 4.5 | 10.6 | 1.35 |
| | | | | | 0.075 | 5 | 9.5 | 1.5 |
| | | | | | 0.1 | 1 | 63.7 | 0.3 |
| | | | | | 0.1 | 2 | 31.8 | 0.6 |
| | | | | | 0.1 | 3 | 21.2 | 0.9 |
| | | | | | 0.1 | 4 | 15.9 | 1.2 |
| | | | | | 0.1 | 5 | 12.7 | 1.5 |
| | | | | | 0.125 | 2 | 39.8 | 0.6 |
| | | | | | 0.125 | 3 | 26.5 | 0.9 |
| | | | | | 0.125 | 4 | 19.9 | 1.2 |



**Table A2.** Test matrix for the prescribed pitch motion cases.

| Pitch - 2.5 m/s, 300 rpm | | | |
|---|---|---|---|
| $\Delta V^*$ | f [Hz] | A [deg] | $f_r$ [-] |
| 0.0325 | 0.5 | 2.11 | 0.24 |
| 0.0325 | 1 | 1.06 | 0.48 |
| 0.0325 | 2 | 0.53 | 0.96 |
| 0.0325 | 3 | 0.35 | 1.44 |
| 0.0325 | 4 | 0.26 | 1.92 |
| 0.0325 | 5 | 0.21 | 2.4 |

| Pitch - 4 m/s, 480 rpm | | | |
|---|---|---|---|
| $\Delta V^*$ | f [Hz] | A [deg] | $f_r$ [-] |
| 0.0375 | 0.5 | 2.25 | 0.15 |
| 0.0375 | 1 | 1.13 | 0.3 |
| 0.0375 | 2 | 0.56 | 0.6 |
| 0.0375 | 3 | 0.38 | 0.9 |
| 0.0375 | 4 | 0.28 | 1.2 |
| 0.0375 | 5 | 0.23 | 1.5 |
| 0.05 | 0.5 | 3.38 | 0.15 |
| 0.05 | 1 | 1.69 | 0.3 |
| 0.05 | 2 | 0.84 | 0.6 |
| 0.05 | 3 | 0.56 | 0.9 |
| 0.05 | 4 | 0.42 | 1.2 |
| 0.05 | 5 | 0.34 | 1.5 |
| 0.075 | 0.5 | 4.5 | 0.15 |
| 0.075 | 1 | 2.25 | 0.3 |
| 0.075 | 2 | 1.13 | 0.6 |
| 0.075 | 3 | 0.75 | 0.9 |
| 0.075 | 3.5 | 0.64 | 1.05 |
| 0.075 | 4 | 0.56 | 1.2 |
| 0.075 | 4.5 | 0.5 | 1.35 |
| 0.075 | 5 | 0.45 | 1.5 |
| 1 | 0.5 | 6.75 | 0.15 |
| 1 | 1 | 3.38 | 0.3 |
| 1 | 2 | 1.69 | 0.6 |
| 1 | 3 | 1.13 | 0.9 |
| 1 | 3.5 | 0.96 | 1.05 |
| 1 | 4 | 0.84 | 1.2 |
| 1 | 4.5 | 0.75 | 1.35 |
| 1 | 5 | 0.68 | 1.5 |

**Table A3.** Test matrix for the prescribed yaw motion cases.

| Yaw - 4 m/s, 480 rpm | | |
|---|---|---|
| f [Hz] | A [deg] | $f_r$ [Hz] |
| 0.5 | 6.05 | 0.15 |
| 1 | 3.03 | 0.3 |
| 2 | 1.51 | 0.6 |
| 3 | 1.01 | 0.9 |
| 4 | 0.76 | 1.2 |
| 5 | 0.61 | 1.5 |
| 0.5 | 9.08 | 0.15 |
| 1 | 4.54 | 0.3 |
| 2 | 2.27 | 0.6 |
| 3 | 1.51 | 0.9 |





## Appendix B: Six Degrees-of-Freedom Prescribed Motion Test Matrix

**Table B1.** FAST simulation parameters utilized to extract the motion over the 6 degrees-of-freedom. The simulations were performed at full-scale and the motions were prescribed at model scale.

| TripleSpar | | | | |
|---|---|---|---|---|
| Load Case | Wind Speed [m/s] | Rotor Speed [rpm] | Hs [m] | Tp [s] |
| LC1 | 7 | 6 | 1.38 | 7 |
| LC2 | 7.1 | 6.04 | 1.67 | 8 |
| LC3 | 10.3 | 8.27 | 2.2 | 8 |
| LC4 | 11.4 | 9.6 | 3.04 | 9.5 |
| LC5 | 11.4 | 9.6 | 4.29 | 10 |
| LC6 | 11.4 | 9.6 | 6.20 | 12.5 |
| LC7 | 11.4 | 9.6 | 8.31 | 12 |

## Appendix C: Quasi-Steady Theory

The surge and pitch harmonic prescribed motion cases are compared to linear quasi-steady models. Usually, these low-fidelity numerical models are utilized for the design of turbine controllers and performing load analysis at an inexpensive computational cost, examples of applications can be seen in Fontanella et al. (2020), Lemmer (2018) and Pegalajar-Jurado et al. (2018). If quasi-steady aerodynamics are assumed, the thrust force can be expressed by:

$$T = \frac{1}{2}\rho A_{rotor} U^2 C_T(\lambda(\omega, U), \beta) \tag{C1}$$

Where $\rho$ is the air density, $A_{rotor}$ is the area of the rotor and $U$ is the wind speed. The thrust coefficient is a function of the blade pitch angle, $\beta$, and the tip-speed ratio, $\lambda$, which is also a function of the rotor speed $\omega$, and the wind speed. When applying a first-order Taylor linearization for a given operation point, the thrust force can be approximated to the following equation:

$$T \approx T_0 + K_{UT}(U - U_0) + K_{\beta T}(\beta - \beta_0) + K_{\omega T}(\omega - \omega_0) \tag{C2}$$

Where $T_0$, $U_0$, $\beta_0$ and $\omega_0$ represents the steady-state values at the specific operation point. $K_{UT}$, $K_{\beta T}$ and $K_{\omega T}$ denotes the partial derivatives of the thrust with respect to wind speed, the pitch angle and the rotor speed, respectively. Considering that for this test campaign, the rotor speed is set as constant and the turbine has a fixed collective pitch angle. Therefore, the Eq. C2 is reduced to:

$$T \approx T_0 + K_{UT}(U - U_0) \tag{C3}$$





Considering that the maximum apparent wind speed at the rotor depends on the harmonic motion that is imposed, the amplitude of the velocity variation, $\Delta V$, and the amplitude of the thrust variation, $\Delta T$, can be defined as:

$$(U - U_0) = \Delta V = 2\pi f A, \qquad\qquad (T - T_0) = \Delta T = \Delta V K_{UT} = 2\pi f A K_{UT} \qquad\qquad \text{(C4)}$$

Where $f_{motion}$ and $A_{motion}$ denote the motion frequency and amplitude respectively. The nondimensional quantities reduced frequency ($f_r$), reduced amplitude ($A_r$) and normalized velocity variation ($\Delta V^*$) are defines as

$$f_r = f D_{rotor}/U_0 \qquad\qquad A_r = A/D_{rotor} \qquad\qquad \Delta V^* = \Delta V/U_0 \qquad\qquad \text{(C5)}$$

where $D$ is the rotor diameter.

*Data availability.* The dataset is accessible upon request to the authors.

*Author contributions.* FT, FN and AV imagined the scope of the work and designed the experimental campaign. FT and FN carried out the tests. FN performed the analyses of the static cases and FT performed the analyses of the dynamic cases. FN performed the literature 345    review. FT prepared the manuscript including contributions from the co-authors. AV was responsible for the supervision of the tests, the interpretation of the results, and the manuscript revisions.

*Competing interests.* The authors declare that they have no conflict of interest.

*Acknowledgements.* The project has received funding from the European Union's Horizon 2020 research and innovation program under grant agreement No. 860737 (STEP4WIND project, step4wind.eu). This research is also partly funded by the Dutch National Research Council 350    (NWO) under the Talent Programme Vidi scheme (project number 19675).



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
