# Peer review of "An experimental study on the aerodynamic loads of a floating offshore wind turbine under imposed motions"

_Wind Energy Science, 2023_

## Referee Comment (RC2)

Review of wes-2023-86

Title: An experimental study on the aerodynamic loads of a floating offshore wind turbine under imposed motions
By: Federico Taruffi, Felipe Novais and Axelle Viré

Short résumé
A small-scale wind turbine is used to investigate experimentally the unsteady aerodynamic response of a floating wind turbine in a setup using a six-DOF hexapod robot.

General comments:
The author's presents very interesting findings, showing increased thrust amplitude at reduced frequencies above 1.2, in the order of 50% higher. The paper is well written and to the point. The experimental setup and subsequent treatment of acquired measured signals is well explained and the setup carefully engineered, a very nice setup. The low Reynolds no. does have an impact as the SD7032 foil shows decreasing aerodynamic performance at below Re<60k. The cases with wave load does show good comparison, although mainly presenting the capability of the setup, as only limited results is shown. The section could be omitted and be part of a larger investigation on FOWT, varying range of parameters and sea-states.

Specific comments:
L185: foil interpolation with Re in FAST? I think better predictions could be found if Re-interpolation is included.
L192: Fig. 6 is nice, but should be put in context of the real turbine, which would pitch out with decreasing TSR, a comment on this should be included.
L197: The uncertainty is stated, is it high or acceptable?
L213: Nice, Ref. to method? How large is the phase error not using the acc. signals?
Fig. 10 trend rather different at 2.5 vs. 4.0m/s, Low Re impact of the foil data.
L222-L226: The set of equations used here could be stated explicitly related to figure 2.
The reduced frequency $k$ seen by 2D foil sections along the blade, as experienced by a harmonic translating 2D foil, could be estimated. The onset of the unsteady aerodynamic regime is at k-values, $k=pi\ f\ c/U$ above 0.05. Could the inboard portion of the rotor blade be estimated that would experience the unsteady regime? 50% of the blade at 5Hz?

Concluding remarks:
The paper is highly recommended for publication.

Ref.:

Wording - figure:
L305: TripelStar - TripelSpar
Tabel A2. Pitch:  DeltaV =1? Should be 0.1…?

---

## Author Comment (AC1)

**Authors' comment**

**Answers to Referee #1**

RC1: 'Comment on wes-2023-86', Anonymous Referee #1, 13 Sep 2023

The paper describes nicely experiments focusing on thrust measurements on a model turbine mimicking floating motions in different degrees of freedom. The paper is well written and the topic is addressing a highly interesting and important aspect of floating wind turbines. Nevertheless, the authors could improve the paper by adding more details and getting a little more in the discussion.

1) In section 2.1 „Wind turbine model" information of the turbine itself like the diameter, hub height etc. are missing. They are later in table 1 and 2 but the reference to these tables are missing.

 The link to the tables will be added to the text in section 2.1.

2) From figure 1 it looks like the experiments are performed in a closed test section. Here the total blockage of the setup in the test section should be added. The presents of the turbine model could impact the velocity measurements based on the Pitot tube that was placed in the test section. What was the distance between the Pitot tube and the turbine?

The test section of the OJF is actually an open section and the configuration is closed loop. The 2.85m x 2.85m nozzle opens into a test chamber of 13m width and 8m height. More info on the wind tunnel facility configuration can be found at https://www.tudelft.nl/lr/organisatie/afdelingen/flow-physics-and-technology/facilities/low-speed-wind-tunnels/open-jet-facility. The pitot tube that measures the wind velocity indicated in the wind tunnel control panel used for the tests is placed inside the nozzle just upstream of the contraction and the appropriate calculations were made to convert the velocity to the testing location downstream the contraction. This is part of the wind tunnel setup itself. In addition and specifically for the present test, the actual wind speed at the testing location at ca. 1m downstream the nozzle end have been evaluated with a portable fan-type anemometer for the main operating conditions and the values were found in exact agreement with the control panel indication. Considering the open section and the rotor diameter of 1.2m, no blockage effect was kept into account. A mention to the open section and to the velocity verification will be added in the text.

3) The authors say that the turbulence intensity in the wind tunnel is below 0.5% up to 1 m behind the nozzle and lower than 2% 6m behind the nozzle. Where was the turbine placed during the experiment and what is the turbulence intensity at the distance? Also, turbulence intensity might not be enough to really identify any structures in the flow especially when operating the wind tunnel at very low velocities. Did the authors measure the incoming wind without the turbine present? Did they determine the spectrum and scales in the flow? An increase from 0.5% TI to about 2% over a distance of 5m ist quite significant.

The test location at which the turbine is placed is at ca. 1m downstream the nozzle end. It was found in previous studies performed on the flow characteristics of this wind tunnel that the TI is around 0.5%. Refer to Lignarolo et al., 2014, "Experimental analysis of the wake of a horizontal-axis windturbine model" and in particular section 2.5. The indication of the turbine positioning inside the test section will be added in the manuscript.

4) The authors say that since the turbine has a fixed pitch there are no problems with any misalignment between the blades — who did they measure it? Even by mounting three blades in a fixed position there can be some differences in the pitch angle.

The rotor is actually composed of a single piece. It was manufactured in carbon fiber using a single mold that includes the three blade and also the hub. Thus no pitch angle mounting error is possible since the blades cannot be mounted or dismounted.

5) It is not quite clear how the turbine is operated. Is the turbine actively driven bei the motor or is there a control system that keeps the rotational frequency constant? The authors should explain that a little more. Could the constant speed of the turbine effect the comparability of the results to a real turbine which has variable speed control?

The rotor drive features an internal control system that keeps the rotational speed of the rotor fixed at the desired value. The gains of this controller have been adjusted to obtain the most desirable behaviour, i.e. a rotational speed as stable as possible also during motion. During operation, after the startup, the motor works as a generator/brake and dissipates the power with braking resistors.

In these tests the effect of the wind turbine control system is not matter of study. To isolate the effect of the variation in relative wind speed faced by the rotor it was preferred to keep the rotational constant instead. We agree that this does not fully represents a realistic condition, however we believe that this is indeed the optimal way to investigate the isolated effect of the relative wind speed variation due to the motion. Future works may see the implementation of a torque controller to study its effect on the loads and its influence on the system dynamics. However, it is believed that the torque control does not have a critical influence on FOWTs dynamics, or at least of an entity not comparable to the pitch control, which however cannot be studied with the present setup.

6) In figure 3, what is the dotted line?

The dotted line represents the maximum velocity/acceleration according to the Hexapod manufacturer specifications, i.e. the nominal limits. This information was missing and it will be added to the text.

7) In section 4.1 in the discussion of figure 4 the authors discuss torque instead of power. They should make clear which relation they use in order to be able to do so. Also, they should mention the uncertainty in the torque measurement.

In this work the power, intended as rotor mechanical power, is generally calculated as torque times rotational speed. The torque is directly measured by the load cell (component Tx) and the motor speed is measured by the motor encoder and converted from high-speed-shaft to low-speed-shaft using the gearbox ratio.

The static values shown are the average of signals of ca. 30s. No further uncertainty analysis was performed on power and torque as they are no taken into consideration in the dynamic analysis.

8) The mentioned uncertainty in the thrust measurements is about 10%. I can imagine that the signals are also suffering from periodic fluctuations due to small imbalances in the turbine. Is that the case and how did the authors deal with that to come up with the 10% uncertainty ? Also, periodic vibrations in the turbine could couple with frequencies from the hexapod. Did the authors make sure that their result of larger variation in the thrust is caused by such interaction?

The mentioned uncertainty actually referred more to the reproduction of the operating point itself, as the dispersion was not caused by uncertainty on the thrust measure itself but rather on the variability on the operating condition. This, as discussed at the end of section 4.2, is believed to have a minor effect on the thrust variation measure, as this is induced by the motion velocity and a variation of the "static" operating point has little effect on it. For example, if the actual wind speed is 3.8m/s or 4.2m/s instead of 4.0m/s for the rated operating point, this has little effect on the motion induced thrust variation since the wind-to-thrust sensitivity doesn't significantly change if evaluated for mean wind 3.8, 4.0 or 4.2.

Thus, this dispersion value was not indicated do describe the uncertainty on thrust variation. A specific statistical uncertainty analysis has been performed lately. An example case was chosen, namely the surge motion case with U=4m/s and DV/U=0.075 which results were shown in Figure 8. The 40 utilizable motion cycles recorded per test have been split into groups of 8 cycles (a tradeoff to have the greater number of groups and enough cycle per group to perform the frequency analysis proposed in section 4.2). The uncertainty is estimated as the standard deviation of the thrust variation of each group and is evaluated for each motion frequency. The uncertainty is likewise estimated also for the quasi-steady prediction as this is based on measurements too. The resulting thrust variation uncertainty is in the range 2.5% to 8.8% for increasing frequency while the quasi-steady prediction is more constant and around 2.5%.

This uncertainty analysis will be included in the manuscript and the results will also be shown in terms of error bars in Figure 8.

9) The authors should add error bars in the deltaT plots.

Refer to the answer of comment 8.

10) The discussion of figure 11 is very brief. By looking at the presented signals I was wondering why for the lower frequency the measured forces for the condition with and without wind are not in phase while they are in phase for the higher frequency. I did not expect that. Can you comment on that?

A more extensive explanation of Figure 11 follows. The discussion will also be enlarged in the manuscript.
The force in x-direction in no-wind condition (blue dashed) is always in phase with the motion. This is expected, as the force measured in this condition is only the inertial component, which has a 180deg phase shift with respect to the acceleration, which has a 180deg phase shift with respect to the position. Thus it is correct and well expected that the no-wind x-direction force measure is exactly in phase with the motion and it is zero for t=0, instant in which the position is zero. If small shift are presents, they are due to small measurements delays, in particular concerning the Hexapod actual position.
The force in wind condition (blue solid) results shifted in phase with respect to the motion, other than greater in amplitude. This is because the inertial and aerodynamic effect are here summed. While the first is in phase with the motion, the second is ideally at -90deg as it depends on velocity. The actual shift is quantitively evaluated in the frequency analysis (see section 4.2) resulting indeed around -90deg for most cases. Thus, the sum of the two effects, i.e. what is measured in these wind cases, results shifted of a phase between 0 and -90deg. The order of magnitude of the shift is roughly given by the ratio between aerodynamic and inertia forces and this explains why for the higher frequencies case in figure b this shift is less visible than in figure a. This is because increasing the frequency and keeping constant the delta velocity, inertial effect increases significantly and the aerodynamic one remains (ideally, unless unsteadiness) constant.

The force difference (red), representing only the aerodynamic contribution, results at ca. -90deg with respect to the motion, as expected (ideally it is -90deg, as mentioned above). This result is obtained for both low frequency and high frequency cases. However, quantitative results are obtained by the analysis in frequency and this analysis in time is only useful for a deeper and clearer understanding of the dynamics behind.

11) In the same figure the force measurement at time 0s for the case with wind is negative. Could the authors explain that.

See answer 10.

12) I do not really see the benefit of adding the pitch motion, yaw motion and wave load case to the paper. All these sections are very short and the conclusions are not quite clear to me. Especially the results shown in figure 13 should be shown with error bars since the variation between the single frequencies are very small.

The authors believe that pitch and yaw cases are still relevant, although the main findings are related to surge cases. In particular, pitch motion can affect the wake in a different way than surge motion, e.g. the wake meandering shown in studies, and this may have an effect also on loads. Yaw cases are wholly different from surge and pitch as the main drive is not the variation in hub relative velocity, which is null, but the dynamic misalignment. For these reasons the authors would rather keep these contents in the article.

Agreeing that the respective sections are not in-depth enough, discussion and conclusions on pitch and yaw cases will be enriched in the manuscript. For an estimation of the uncertainty it is possible to refer to the uncertainty analysis outcome discussed in the answer of comment 8.

Regarding the wave cases section, instead, the authors agreed to entirely omit this section and utilize the findings for future studies. As this was indeed showing the capabilities of this setup and it was preparatory for the development of a hardware-in-the-loop setup, a short mention will be inserted in the manuscript.

13) The frequency in figure 14 is not clear to me. The real turbine case should move at very slow frequencies compared to the model turbine. From the description in the text it is not clear why they are in the same range. The authors should also discuss the sharp peak int he experimental data. Since the frequency is not the same for the two cases I wonder what the origin is.

To ease the comparison, everything is reported at full scale, as mentioned in the figure label. The experimental results were thus up-scaled to full scale. For clarity, a frequency of 5Hz at model scale corresponds to ca. 0.1Hz at full scale. The scaling factor of frequency (full/model) can be calculated as l_freq=1/l_time=l_velocity/l_length=3/148.

The peak represents the rotation frequency of the rotor. The value is not the same for both cases since cases LC3 and LC4 have different rotor speed (see table B1 in Appendix).

**Answers to Referee #2**

RC2:

Short résumé:
A small-scale wind turbine is used to investigate experimentally the unsteady aerodynamic response of a floating wind turbine in a setup using a six-DOF hexapod robot.

General comments:
The author's presents very interesting findings, showing increased thrust amplitude at reduced frequencies above 1.2, in the order of 50% higher. The paper is well written and to the point. The experimental setup and subsequent treatment of acquired measured signals is well explained and the setup carefully engineered, a very nice setup.

1) The low Reynolds no. does have an impact as the SD7032 foil shows decreasing aerodynamic performance at below Re<60k.

The SD7032 airfoil is more suitable for low-Re application than the DTU 10 MW airfoils, but its performance is indeed decreasing for lower Re numbers. This affects more the below-rated operating point (U = 2.5m/s), as the results reflect. To provide some number, for the rated operating point (U = 4) the maximum Re experienced by an airfoil is in the order of 50k, while for the below rated point it decreases to 30k. For more details refer to the cited work about the aerodynamic design of this rotor (Fontanella et al., 2023). A sentence will be added in the manuscript to account for this in the results explanations.

2) The cases with wave load does show good comparison, although mainly presenting the capability of the setup, as only limited results is shown. The section could be omitted and be part of a larger investigation on FOWT, varying range of parameters and sea-states.

The authors agreed to entirely omit this section and utilize the findings for future studies. As this was indeed showing the capabilities of this setup and it was preparatory for the development of a hardware-in-the-loop setup, a short mention will be inserted in the manuscript.

3) Specific comments:
L185: foil interpolation with Re in FAST? I think better predictions could be found if Re-interpolation is included.

Re-interpolation was active in FAST AeroDyn module for the accounted simulations.

4) L192: Fig. 6 is nice, but should be put in context of the real turbine, which would pitch out with decreasing TSR, a comment on this should be included.

This figure will be omitted as it may be redundant for the scope of the work. The Cp and Ct values relevant for the tests are already shown in Figure 5.

5) L197: The uncertainty is stated, is it high or acceptable?

A statistical uncertainty analysis has been performed lately for an example case (see answer to referee #1 comment 8). The outcome will be included in the paper also in the form of error bars in Figure 8.

6) L213: Nice, Ref. to method? How large is the phase error not using the acc. signals?

The method does not actually have a reference as it has been developed ad-hoc and it's entirely reported in the manuscript. In addition, the answer to referee #1 comment 10 goes deeper in the physical explanation of the signals.

Not using the acceleration signal but simply synchronizing in time the motion time history of no-wind and wind tests not only leads to an error in the evaluation of the phase, but foremost it leads to an error in the evaluation of the thrust variation magnitude itself. Indeed, any phase shift in the force measurements between the no-wind and the wind tests that is given by synchronization errors has a big impact on the variation magnitude itself. Using the tower-base acceleration as common signal and referring every other signal to it allowed to minimize this error as this signal was found to have, on one side, less acausal delays than the motion feedback coming from the hexapod, and on the other end is not influenced by any tower flexibility. Previous draft results without this method were less consistent.

7) Fig. 10 trend rather different at 2.5 vs. 4.0m/s, Low Re impact of the foil data.

Refer to the answer above regarding Re values. This comment will be included in the manuscript.

8) L222-L226: The set of equations used here could be stated explicitly related to figure 2.

The key equation referring to the "adjusted" quasi-static thrust variation will be added in the text after those lines.

9) The reduced frequency *k* seen by 2D foil sections along the blade, as experienced by a harmonic translating 2D foil, could be estimated. The onset of the unsteady aerodynamic regime is at k-values, *k=pi f c/U* above 0.05. Could the inboard portion of the rotor blade be estimated that would experience the unsteady regime? 50% of the blade at 5Hz?

The reduced frequency k=pi*f*c/V has been calculated for the extreme motion cases (f=1Hz and f=5Hz) and the rated operating point (U=4m/s) along the blade span. Indeed, k is greater than 0.05 for ca. the 45% of the blade, starting from the hub side. A remark on this will be added in the text.

Concluding remarks:
The paper is highly recommended for publication.

Ref.:

Wording - figure:

10) L305: TripelStar – TripelSpar

It will be corrected in the text.

11) Tabel A2. Pitch: DeltaV =1? Should be 0.1...?

It will be corrected in the text.